# The Extract of *Perilla frutescens* Seed Residue Attenuated the Progression of Aberrant Crypt Foci in Rat Colon by Reducing Inflammatory Processes and Altered Gut Microbiota

**DOI:** 10.3390/foods12050988

**Published:** 2023-02-26

**Authors:** Weerachai Chantana, Rentong Hu, Songphon Buddhasiri, Parameth Thiennimitr, Payungsak Tantipaiboonwong, Teera Chewonarin

**Affiliations:** 1Department of Biochemistry, Faculty of Medicine, Chiang Mai University, Chiang Mai 50200, Thailand; 2Department of Laboratory Medicine, The Affiliated Hospital of Youjiang Medical, Baise 533099, China; 3Department of Veterinary Biosciences and Veterinary Public Health, Faculty of Veterinary Medicine, Chiang Mai University, Chiang Mai 50200, Thailand; 4Department of Microbiology, Faculty of Medicine, Chiang Mai University, Chiang Mai 50200, Thailand; 5Division of Biochemistry and Nutrition, School of Medical Sciences, University of Phayao, Phayao 56000, Thailand

**Keywords:** aberrant crypt foci, colon cancer prevention, inflammation, *Perilla frutescens* seed residue

## Abstract

*Perilla frutescens* (PF) seed residue is a waste from perilla oil production that still contains nutrients and phytochemicals. This study aimed to investigate the chemoprotective action of PF seed residue crude ethanolic extract (PCE) on the inflammatory-induced promotion stage of rat colon carcinogenesis and cell culture models. PCE 0.1 and 1 g/kg body weight were administered by oral gavage to rats after receiving dimethylhydrazine (DMH) with one week of dextran sulfate sodium (DSS) supplementation. PCE at high dose exhibited a reduction in aberrant crypt foci (ACF) number (66.46%) and decreased pro-inflammatory cytokines compared to the DMH + DSS group (*p* < 0.01). Additionally, PCE could either modulate the inflammation induced in murine macrophage cells by bacterial toxins or suppress the proliferation of cancer cell lines, which was induced by the inflammatory process. These results demonstrate that the active components in PF seed residue showed a preventive effect on the aberrant colonic epithelial cell progression by modulating inflammatory microenvironments from the infiltrated macrophage or inflammatory response of aberrant cells. Moreover, consumption of PCE could alter rat microbiota, which might be related to health benefits. However, the mechanisms of PCE on the microbiota, which are related to inflammation and inflammatory-induced colon cancer progression, need to be further investigated.

## 1. Introduction

An inflammatory condition is a significant contributor to colon carcinogenesis’s onset and promotion [1]. Patients with inflammatory bowel disease (IBD) who have a family history of colorectal cancer have a higher risk of developing colon cancer [2]. In the tumor microenvironment, inflammation occurs due to the presence of immune cells that are in charge of clearing aberrant cells [3,4]. On the other hand, the presence of inflammatory cytokines in the tumor environment causes cell proliferation, resistance to cell death, tumor invasion, and metastasis [5,6,7,8]. Moreover, the gut microbiome is a key factor in inflammation-related multistage colon carcinogenesis [9]. The increase in beneficial colonic bacteria such as *Lactobacillus* and *Bifidobacterium* could reduce the incidence of colon carcinogenesis [10]. Therefore, controlling inflammation and modifying the gut microbiome in the colon by consuming natural products could modulate the progression of colon cancer.

*Perilla frutescens* (Nga-kee-mon; Thai word) is a plant that is mostly found in Asian countries such as Korea, Japan, China, and Thailand [11]. Many studies have demonstrated that PF has antioxidant, anti-inflammation, and anti-cancer biological activities [12,13,14]. PF leaves contain high levels of bioactive compounds, including rosmarinic acid, apigenin, and luteolin [15]. In previous studies, PF leaf extraction showed an ability to inhibit the growth of colon cancer cell lines and suppress rat colon cancer progression [15,16]. Although the biological activity of Perilla seed oil has been widely reported [17,18,19], the activity of seed residue, a waste product from PF seed oil production, is limited. Therefore, PF seed residue is an interesting part of studying bioactive compounds, which have colon cancer-preventive properties.

This study aimed to investigate the inhibitory activities of an ethanolic extract of *Perilla frutescens* seed residue on the promotion stage of rat colon carcinogenesis, general and local gut inflammation, and alteration of the gut microbiome. Moreover, the molecular mechanisms of PCE on inflammatory responses in both macrophage and human colon cancer cell lines were also determined.

## 2. Materials and Methods

### 2.1. Plant Material and Extraction

The seed residue used for this study was taken from perilla seed oil production in Baan San Khong, Dok Kham Tai district, Phayao, Thailand. The voucher number of plant material was deposited at the Queen Sirikit Botanic Garden Herbarium, Chiang Mai, Thailand (Code: QBG-93756). Dried PF seed residue was extracted with 70% ethanol (1:10 *w*/*v*) overnight at room temperature. After filtration, a concentrated extract was lyophilized to obtain PF seed residue crude ethanol extract (PCE) powder. All extracts were kept at −20 °C for further study.

### 2.2. Determination of Phenolic Compound

Total phenolic contents were measured using the Folin–Ciocalteu reagent assay [20] and presented in milligrams of gallic acid equivalent per gram of extract. Subsequently, flavonoid contents were measured using an aluminum chloride colorimetric assay [21] and presented in milligrams of catechin equivalent per gram of extract.

### 2.3. HPLC Fingerprint Analysis

PCE were subjected to determine known phenolic acids by HPLC, according to the report of Pintha et al. [22]. All samples were filtered and then separated by ODS-3-C18 column (4.6 × 250 mm, 5 µm particle diameters) (Agilent, Santa Clara, CA, USA). Mobile phases A and B were 0.1% trifluoroacetic acid and methanol, respectively. Separation and elution were performed as follows: 0–35 min, 90–10% mobile phase A, and 10–90% mobile phase B; 35–40 min, 10–90% mobile phase A, and 90–10% mobile phase B, with a flow rate of 1.0 mL/min. A UV detector at wavelength 320 nm was used for the measurement of phenolic content peaks. The concentration of compounds presented in the HPLC profile was calculated by phenolic acid standard curves.

### 2.4. Animal Model

Male Wistar rats, aged four weeks, weighing 60–90 g, were obtained from the Siam Nomura. International Co., Ltd., Bangkok Thailand. The animal experiment protocol was authorized by the animal ethics committees at the Faculty of Medicine, Chiang Mai University, Chiang Mai, Thailand (Protocol No: 26/2563, approved date: 7 October 2020). The rats were kept in air conditioning set to 25 °C with a 12:12 h light/dark cycle. Rats were given a standard laboratory pellet diet and sufficient amounts of water.

### 2.5. Inhibitory Activities of PCE on Aberrant Crypt Foci Progression

After 1 week of acclimatization, rats were randomly separated into five groups of six rats with comparable average weights (111.87 ± 2.09 g). In control groups (groups 1 and 5), rats received a subcutaneous (*s.c.*) injection of 0.9% normal saline once a week for two weeks and were fed with 10% DMSO (group 1; negative control) or 1 g/kg body weight of PCE (group 5; PCE control) in weeks five to 15. In positive control (group 2) and experimental groups (groups 3 and 4), rats were *s.c.* injected with 40 mg/kg body weight of dimethylhydrazine (DMH) at week 1 and week 2. At week 3, rats were then given 1% dextran sulfate sodium (DSS) daily instead of drinking water for one week. In week 5, rats were orally fed with 10% DMSO (group 2) and 0.1 or 1 g/kg body weight/day of PCE (groups 3 or 4, respectively) until week 15. Body weights were recorded once a week. Blood was collected from the lateral tail vein of the rats in week 3 (after DMH administration), week 5 (after DSS-induced inflammation), and week 10 (after five weeks of PCE intervention). The serum was subjected for the examination of pro-inflammatory cytokine concentration (IL-6, IL-1β, TNF-α) by ELISA kit (Thermo Fisher Scientific, Waltham, MA, USA ) following manufacturer’s instructions protocol. Fecal specimens were naturally collected (week 14, non-invasive, and sampled repeatedly) and subjected to intestinal microbiota studies. At the end of the experiments, all rats were sacrificed, and colons were collected for ACF evaluation (Figure 1A).

### 2.6. Aberrant Crypt Foci Analysis

Each rat colon was expanded by 10% formaldehyde in PBS (pH 7.4) and placed on ice for at least 30 min. After that, each colon was opened and divided into the rectum (2 cm from the anus), proximal segment, and distal segment. Then, colons were flattened between filter papers and kept in 10% formaldehyde in PBS for at least 24 h. Each piece of colon was stained with 2% methylene blue for 1–2 min. Under a light microscope, the quantity and size of ACF were graded in accordance with the Bird RP criteria [23,24]. Compared to normal crypts, aberrant crypts were bigger and had a thicker epithelial lining, and usually gathered into a focus of small (1–3 AC/f) or clusters of abnormally large crypts (>4 AC/f).

### 2.7. Gut Microbiome Analysis

Bacterial genomic DNA was extracted from the feces of each group of rats using the TIANamp Stool DNA Kit (Tiangen, Beijing, China). The bacterial 16S rRNA genes V4 region was amplified by polymerase chain reaction (PCR) using 515F (5’- GTGCCAGCMGCCGCGGTAA-3’) and 806R (5’- GGACTACHVGGGTWTCTAAT-3’) primers. Then, paired-end 2 × 250 bp sequencing was performed by the Illumina NovaSeq 6000 platform. Then, the Quantitative Insights Into Microbial Ecology 2 version 2022.8 (QIIME2) pipeline was used to analyze the sequence reads. The paired-end sequences were de-noised and merged using the DADA2 plugin within QIIME2. Taxonomic assignment of 16S rRNA sequences was performed using a Silva 138 99% taxonomy classifier [25,26]. Amplicon sequence variants (ASVs) were aligned using the mafft plugin in QIIME2. Shannon and Simpson indices were assessed for alpha diversity. Beta diversity was analyzed using the Bray-Curtis distance matrix and visualized by principal coordinate analysis (PCoA) in R software v4.0.1. Differential abundance analysis was assessed using linear discriminant analysis (LDA) effect size (LEfSe) in the Galaxy module (http://huttenhower.sph.harvard.edu/galaxy, accessed on 12 October 2022) [27] to identify significant bacterial taxa among groups. The relative abundances of the taxa were compared. Microbiome data were analyzed using the Kruskal–Wallis test with multiple comparisons and the Wilcoxon rank-sum test.

### 2.8. Induction of Short-Term Colonic Inflammation and PCE Administration

Rats were randomly divided into five groups with four rats of similar average weights. The induction was performed in a similar way to the inhibitory activities of PCE in the ACF progression experiment. In week 4, rats were orally fed with 10% DMSO (groups 1 and 2) and 0.1 or 1 g/kg body weight of PCE (groups 3, 4, or 5, respectively). Body weights were recorded once a week. All rats were sacrificed at the indicated time, and then colon tissue was collected for the determination of pro-inflammatory cytokine mRNA levels (Figure 1B).

### 2.9. Evaluation of Inflammatory Related Cytokines

Total RNA from scraped colonic mucosal cells was reverse transcribed to cDNA by the ReverTra Ace^®^ qPCR RT kit (TOYOBO, Osaka, Japan), in line with the manufacturer’s instructions. Quantitative PCR was carried out using the Maxima SYBR Green qPCR Kit (Thermo Fisher Scientific; Waltham, MA, USA) and the ABI 7500 real-time PCR system. Primer sequences are listed as follows: TNF-α: 5′-TTCTCATTCCTGCTCGTGGC-3′(forward) and 5′-AACTGATGAGAGGGAGCCCA-3′ (reverse), IL-6: 5′-TCCTACCCCAACTTCAATGCTC-3′ (forward) and 5′-TTGGATGGTCTTGGTCCTTAGCC-3′ (reverse), IL-1β: 5′-CACCTCTCAAGCAGAGCACAG-3′ (forward) and 5′-GGGTTCCATGGTGAAGTCAAC-3′ (reverse). The set-up temperature and time are as follows: first denaturing for 10 min, followed by 40 cycles of 95 °C for 15 sec and 60 °C for 1 min. Observations of each gene were expressed compared to control cells after being adjusted to GAPDH.

### 2.10. Cell Lines and Culturing

HCT-116 and HT-29 (human colorectal cancer cell lines) and RAW 264.7 (a murine macrophage cell line) were purchased from the American Type Culture Collection (ATCC, Manassas, VA, USA). The cells were cultured in Dulbecco’s Modified Eagle’s Medium (DMEM) with 10% fetal bovine serum (FBS), 100 units/mL penicillin, and 100 units/mL streptomycin at 37 °C in an incubator supplied with 5% CO_2_.

### 2.11. Inflammation-Induced Cell Proliferation

Briefly, 2.5 × 10^3^ cells/well of HCT-116 and HT-29 were cultured in serum-free media overnight. Then, 10 ng/mL of IL-6 was added, and plates were incubated for 24 h in an incubator. Afterward, 0–200 μg/mL of PCE were added and incubated for 0, 24, and 48 h. Cell viability was assessed by the MTT test at the set period compared to 100% cell viability at 0 h and calculated as relative cell growth.

### 2.12. Inflammatory Cytokine Production

In 6-well plates, 5 × 10^5^ cells/well of RAW 264.7 cells were cultured in DMEM with 1 µg/mL of LPS for 24 h. After that, cells were rinsed twice with PBS and continuously cultured with various concentrations of PCE for 24 h. Finally, a cell culture media was subjected to an evaluation of the concentration of IL-6, IL-1β, and TNF-α by ELISA kit (Thermo Fisher Scientific, Waltham, MA, USA) following manufacturer’s instructions protocol.

### 2.13. Statistical Analysis

The data from in vitro and in vivo experiments were presented as the mean ± standard deviation (SD) using Microsoft Excel. The statistical analysis for significant differences among all the data was conducted by one-way ANOVA (GraphPad Prism software, version 9.0.0 (121); San Diego, CA, USA). The significance of the difference or correlation between experimental groups was accepted at *p* < 0.05 and *p* < 0.01.

## 3. Results

### 3.1. The Extraction Yield of Perilla Seed Residue

One kilogram of perilla seed residue yielded 73.40 ± 6.50 g of lyophilized powder (7.34 ± 0.65 g%). The average of phenolic compounds was 49.65 ± 3.56 mg of gallic acid equivalents/gram extract, while the flavonoid content was 41.02 ± 2.68 mg of catechin equivalents/gram extract. Additionally, the concentrations of rosmarinic acid, luteolin, and apigenin were 21.06 ± 2.13, 11.11 ± 0.40, and 6.55 ± 1.78 mg in 1 g of PCE, respectively.

### 3.2. Effect of PCE on DMH and DSS Induced Rat ACF Progression

The mean numbers of ACF and crypt/focus of experimental rats with percentages of inhibition are shown in Table 1. The ACF was not observed in the saline-treated group, while all rats administered by DMH exhibited ACF, which was distributed throughout the colon with a range of one to more than eight aberrant crypts per focus). The mean number of ACF in the positive control group was 388 ± 90.11 ACF/rat. Rats fed with 1.0 g/kg body weight of PCE after DMH and DSS administration for 10 weeks showed 66.46% lower mean number of total ACF (130 ± 31.78 ACF/rat) than DMH + DSS treated alone (*p* < 0.01). The inhibitory effects of PCE were strongly observed in the proximal part since the number of ACF was decreased to 77.07%, compared to the positive control group (*p* < 0.01). Moreover, the AC/f slightly decreased to 2.81 ± 0.40 AC/f in the low dose (15.12%) and to 3.09 ± 0.27 in the high dose (6.69%) of the PCE feeding group, respectively; nevertheless, the significant difference to the DMH + DSS treated group (3.31 ± 0.41 AC/f) was not observed. These results suggested that PCE could reduce the progression of ACF to a large amount and recover the aberrant crypt to normal crypt, which showed a lower number of ACF.

### 3.3. The Effect of PCE on the Level of Srum Pro-Inflammatory Cytokines

The levels of cytokines, including IL-6, IL-1β, and TNF-α in rat serum, are shown in Figure 2A–C. At week 3, rats that received DMH resulted in significantly increased serum levels of IL-6 (1181.2 ± 0.8 pg/mL), IL-1β (668.4 ± 0.9 pg/mL), and TNF-α (241.8 ± 0.5 pg/mL) more than 2–3.5-fold when compared to the negative control group (770.7 ± 0.4, 343.7 ± 0.9 and 118.4 ± 0.3 pg/mL, respectively) (*p* < 0.05). At week 5, after one week’s DSS administration, the serum levels of these three cytokines were higher than in week 3, which was due to the inflammation in rat colons caused by DSS. In week 10, after five weeks’ oral gavage with 0.1 and 1 g/kg body weight of PCE, the serum levels of all cytokines were significantly lower (*p* < 0.05) than in the positive control group. This result demonstrated that DMH and DSS treatments induced the inflammatory process and were related to the promotion of colon carcinogenesis. After rats were fed with PCE, extenuated inflammatory cytokine production occurred, which could reduce pro-inflammatory cytokine production. The distribution of ACF from 1 to more than 8 AC/f throughout the colon was shown in Figure 3A. Moreover, the correlation between pro-inflammatory cytokines and ACF/rat was shown in Figure 3B, C. Taken together, PCE could suppress the mRNA expression of pro-inflammatory cytokines in colonic epithelial cells and reduce systemic cytokine production, which reduced aberrant colonic epithelial cell progression in rats.

### 3.4. The Effect of PCE on the Expresion of Inflammatory Cytokine in Colonic Epithelial Cells

The fold change of mRNA expression is shown in Figure 2D–F. In the DMH + DSS group, a significant increase in the levels of IL-6, IL-1β, and TNF-α mRNA was observed (1.70 ± 0.16, 1.59 ± 0.00, and 2.08 ± 0.34-fold, respectively), compared to a negative control group. The mRNA levels of IL-6 and TNF-α in colonic epithelial cells were significantly decreased (1.23 ± 0.00, 1.29 ± 0.08-fold, respectively) (*p* < 0.05) in rats receiving DMH + DSS with a high dose of PCE (1 g/kg body weight), while the mRNA level of IL-1β was significantly decreased in both the low and high dose treatment of PCE (1.28 ± 0.07, 1.08 ± 0.02-fold, respectively). As with the serum cytokine levels, PCE had no effect on the expression of cytokines in colonic epithelial cells of untreated animals. The results suggested that PCE might have the potential to extenuate the inflammatory process by down-regulating pro-inflammatory cytokines, and reduced cytokines promoted aberrant crypt progression in rat colons.

### 3.5. Altered Rat Gut Microbiota Composition by Reduction of the Dysbiotic Gut Marker in Rats

Firstly, PCE increased the dose-dependent gut microbiota diversity in the CRC rat model. In rats receiving DMH + DSS, gut microbiota diversity within the sample (alpha diversity) was significantly reduced compared to that of the healthy control group (Shannon entropy and Simpson’s diversity for Figure 4A,B, respectively). Interestingly, the rat’s gut microbiota beta diversity with Bray-Curtis dissimilarity could be observed only in rats fed with the high dose of PCE (1 g/kg rat body weight), not a low dose (0.1 g/kg rat body weight) (Figure 4C).

Next, the relative abundance of bacterial biomarkers was determined and is expressed in Figure 5. There was no significant difference in the *Firmicutes/Bacteroidota* (F/B) ratio in any of the groups of rats. Interestingly, there was an increase in the relative abundance of *Gammaproteobacteria*, an important class of bacteria in the phylum *Pseudomonadota* (formerly named *Proteobacteria*) in DMH + DSS-treated rats (Figure 5B), which have been considered an important marker for gut dysbiosis in several medical conditions, including colorectal cancer [28].

On the other hand, consumption of PCE reduced the increase in *Escherichia/Shigella*, the representative bacteria belonging to a class of *Gammaproteobacteria* (Figure 5C). These data suggested that PCE consumption may attenuate the chronic inflammatory condition in the gut of DMH + DSS-treated rats and reduce *Gammaproteobacteria* levels. PCE also decreased the Gram-negative *Paraprevotella* (Phylum *Bacteroidota*) in the rat (Figure 5D). Interestingly, PCE increased the relative abundance of beneficial bacteria such as *Muribaculaceae*, *Lactobacillus*, and *Oscillospiraceae* (Figure 5E–G, respectively).

### 3.6. The Effect of PCE on Cell Proliferation of Colon Cancer Cell Lines

The relative cell growth of HCT-116 and HT-29 at 24 and 48 h, compared with the starting time, is shown in Figure 6A,B. The treatment with IL-6 at 10 ng/mL resulted in a significant increase in the relative growth of HCT-116 and HT-29 at both 24 and 48 h when compared with a negative control group. Therefore, the growth rate of both types of colon cancer cells was increased in the presence of IL-6.

When colon cancer cells were treated with PCE in the presence of IL-6, PCE at 100 and 200 µg/mL significantly inhibited the growth of both HCT-116 (about 12% (*p* < 0.05) and 25% (*p* < 0.01), respectively), and HT-29 (about 12% (*p* < 0.05) and 24% (*p* < 0.01), respectively), after 48 h, when compared to the positive control groups (Figure 6C,D). As expected, PCE alone did not affect the growth of both HCT-116 and HT-29. This result confirmed that PCE showed anti-proliferative properties on colon cancer cells in an inflammatory condition.

### 3.7. The Effect of PCE on the Secretion of Pro-Inflammatory Cytokines

The concentration of IL-6, IL-1β, and TNF-α was measured in the cell culture media of LPS-activated RAW 264.7 cells by an ELISA kit. The percentages of all cytokines produced are shown in Figure 7A–C. When RAW 264.7 cells were treated with 1 µg/mL of LPS resulted in a significant, more than 2–4 fold, increase in IL-6 (280.45 ± 9.09 pg/mL), IL-1β (284.57 ± 76.07 pg/mL), and TNF-α (758.36 ± 25.88 pg/mL), compared to the negative control (90.52 ± 3.74, 65.11 ± 11.48, and 345.21 ± 29.61 pg/mL, respectively) (*p* < 0.01). After 24 h of culturing with PCE (100 and 200 µg/mL), the concentration of cytokines in cell culture media significantly decreased to 178.91 ± 1.39 pg/mL (IL-6), 132.91 ± 1.15 pg/mL (IL-1β), and 88.56 ± 0.60 pg/mL (TNF-α) (80–200%, *p* < 0.01) compared to the positive control. The results demonstrated that PCE attenuated the inflammatory process by reducing the cytokine production in LPS-induced RAW 264.7 cells.

## 4. Discussion

The seed of *Perilla frutescens* is a great source of nutrients, especially essential fatty acids [22]. The biological activities of PF seed oil, including its anti-atherosclerosis and neuroprotective effects, are widely promoted [29,30]. Therefore, PF seed residue from PF oil production that has been used for animal feeds remains largely useless. Recently, it has been reported that PF seed residue contains a high amount of nutrients and phytochemicals [31]. After perilla seed oil production, the remaining 72.5% of PF seed residue still contains nutrients and bioactive compounds [19,31]. To increase the value of perilla waste product, its active compounds and biological activities were investigated in this study. The results revealed that the extract of PF seed residue (PCE) presented the same level of phenolic and flavonoid compounds, which were similar to the leaf extract of PF, but the concentration profile was different. HPLC analysis found 21.06 mg of rosmarinic acid, 11.11 mg of luteolin, and 6.55 mg of apigenin in one gram of PCE, compared to 148.0, 0.9, and 0.4 mg/g, respectively, in the leaf extract [15]. It was deduced that the residue of seed and leaves from PF, which contained the difference in the phytochemical ratio, might show biological effects that are similar or different. The effective dose of PCE was referred to our previous research, which suggests that a crude ethanolic extract of PF leaves containing high rosmarinic acid exhibited anti-inflammation in vitro and inhibitory effects on DMH-induced ACF formation in rats in the promotion stage [15]. Moreover, the crude ethanol extract and its partially purified fraction suppressed the growth of colon cancer cells through the alteration of the inflammatory signaling pathway [32]. Therefore, we hypothesized that PF seed residue may have the same properties as its leaves, especially anti-cancer activity.

This study has firstly demonstrated the chemopreventive effect of PF seed residue on the progression of rat colon carcinogenesis, which had been induced by chronic inflammation. The inflammation-induced progression of aberrant crypt foci in the rat colon was used to determine the efficiency of PF seed residue. ACF growth is generally inhibited by inducing cell cycle arrest and apoptosis induction in colon epithelial cells [33] or by inflammatory regulation, which is the target in colon cancer prevention [34]. Chemoprevention can exert anti-inflammatory activity for applications in cancer prevention. Rosmarinic acid (containing seven metabolites such as trans-caffeic acid and trans-m-coumaric acid), a polyphenol compound that is discovered in many plants, especially *P. frutescens*, luteolin (containing eight metabolites such as luteolin-3′-O-β-d-glucuronide), and apigenin (whose major metabolite is luteolin) have been shown to possess anti-inflammatory properties, such as lowering TNF-α, IL-6, and IL-1β production or downregulating iNOS, via modulation of HMGB1/TLR4, NF-kB, AP-1, and TGF-β1/SMAD pathways [35,36,37]. Therefore, in this study, we investigated the effect of perilla seed residue extract on DSS-promoted ACF progression in DMH-treated rat colons. High-dose PCE strongly reduced the mean number of ACF, compared to the DMH/DSS-treated group (66.46%); while low-dose PCE slightly decreased the number of AC/focus (15.12%). It has been reported that DMH induces genetic alterations in colonic epithelial cells and that ACF is formed. Then, DSS treatment alters the colonic mucosal inflammation from various cytokines, leading to ACF progression [38,39]. Moreover, DMH/DSS promotes the expression of iNOS, COX-2, TNF-, and IL-1 in rat colonic mucosal cells [34]. Therefore, it could be summarized that PCE feeding after receiving DMH and DSS could suppress the progression of aberrant colonic epithelial cells. Next, we hypothesized that PCE might modulate the inflammatory environment in the rat lumen. Therefore, the inflammatory process, which is related to aberrant colonic epithelial cell progression, was investigated.

The effect of PCE on the serum level of pro-inflammatory cytokines, consisting of IL-6, IL-1β, and TNF-α in rat serum, was determined by ELISA. The results revealed that DMH resulted in an increase in pro-inflammatory cytokine production in rat serum, which might be caused by colonic epithelial cell mutation and aberration of immune cell infiltration [40]. After rats were induced to develop colitis by DSS administration for one week, an increase in pro-inflammatory cytokine production occurred, leading to the promotion of aberrant colonic cell progression. By contrast, five weeks of PCE feeding in DMH + DSS rats resulted in a decrease in inflammatory cytokine production in a dose-dependent manner. Furthermore, an analysis of the effects of PCE on IL-6, IL-1β, and TNF-α expression in colonic epithelial cells of the rat colon carcinogenesis model showed an increase in IL-6, IL-1β, and TNF-α mRNA levels in the DMH + DSS group. When rats were fed with PCE after DMH and DSS administration for one week, the mRNA expression of IL-6 and TNF-α in colonic epithelial cells was significantly decreased in high-dose treatment of PCE, while IL-1β expression was significantly decreased in both low- and high-dose treatments of PCE. Consequently, these results suggested that PCE could suppress the expression of pro-inflammatory cytokines, leading to a reduction in aberrant colonic epithelial cell progression in rats.

The inflammatory condition is present in the tumor microenvironment in many types of cancer, including colon cancer. It has been reported that the reduction in the inflammatory microenvironment can modulate the growth of cancer [41,42,43]. Many factors are related to the inflammation in the colonic lumen, which can be targeted with approaches to suppress tumor-promoting inflammation, such as the immune system, tumor metabolism, and gut microbiota [44]. An imbalance of gut microbiota is also a factor that leads to increased inflammation in the colonic lumen. Therefore, the alteration of the bacterial profile in the PCE-treated experimental rat was determined. It was found that rat gut microbiota diversity within the sample (alpha diversity) was significantly reduced in the model of DMH + DSS, compared to that of the healthy control group. Our findings showed that PCE increased the gut microbiota diversity in the CRC rat model. Moreover, there was an increase in the relative abundance of *Gammaproteobacteria*, an important class of bacteria in the phylum *Pseudomonadota* (formerly named *Proteobacteria*) in DMH + DSS-treated rats. High numbers of *Proteobacteria* have been considered an important marker for gut dysbiosis in several medical conditions, including colorectal cancer [28]. Gammaproteobacteria have a lipopolysaccharide outer membrane that causes chronic inflammation through toll-like receptor (TLR)-4 activation in the gut, which can be modulated by some bioactive compounds [45]. Interestingly, PCE reduced the increase in *Escherichia/Shigella*, the representative bacteria belonging to a class of *Gammaproteobacteria*. These findings suggested that PCE consumption might have the ability to attenuate the chronic inflammatory condition in the gut of DMH + DSS rats and reduce *Gammaproteobacteria* levels. Treatment of PCE also increased the relative abundance of beneficial bacteria such as *Muribaculaceae*, *Lactobacillus,* and *Oscillospiraceae*. The protective effects of *Lactobacillus* on the development of precancerous growths and colorectal carcinogenesis in the rat model have been revealed [46]. Moreover, *Ruminococcaceae* (*Oscillospiraceae*) is a family of strictly anaerobic bacteria normally present in the colonic mucosal biofilm of healthy individuals [47]. However, the interaction of gut microbiota, immune cells, inflammatory cytokines, and the response of aberrant colonic epithelial cells is the focal point for the prevention of inflammation-related colon carcinogenesis.

One of the key factors in tumor microenvironments is the activated macrophage, which produces various inflammatory mediators to promote tumor development [48]. In addition, the production of an inflammatory microenvironment by aberrant colonic epithelial cells is also a crucial factor for carcinogenesis [1]. Therefore, the effect of PCE on inflammatory processes in LPS-activated macrophages was investigated. LPS, a major glycolipid presented on the outer membrane of Gram-negative bacteria, potently generates inflammation by activating and infiltrating immune system cells. It could act together with TLR4 (toll-like receptor 4) on the membrane of macrophages, which results in the triggering of MyD88 and TRIF pathways, leading to an increase in TNF-α and IL-6 production via NF-κB and/or MAPK signaling cascades [49]. Our results demonstrated that treatment with PCE could reduce cytokine secretion from LPS-activated RAW 264.7 cells. This model correlated with the inflammation observed in previous animal experiments, where PCE acted effectively against the inflammatory process. As a result, the suppression of aberrant crypt progression in rats is caused by the inhibition of pro-inflammatory cytokine production in macrophage cells. It has been reported that apigenin, luteolin, and other polyphenol compounds can inhibit cytokine production by suppressing the activation of macrophage cells via inhibiting NF-κB, MAPKs, especially the PI3K/Akt signaling pathway [50]. It is well documented that the presence of cytokines such as IL-1, IL-6, and TNF-α or chemokines such as CCL2 and CXL8 from white blood cells or tumor-associated macrophage (TAMs) generate cancer-related inflammation in tumors [51]. Therefore, this result indicated that the reduction in cytokine expression and/or secretion from macrophages might lessen the tumor’s inflammatory reaction.

In the microenvironment, tumor cells also produce inflammatory cytokines and prostaglandins via coordination of transcription factors including STAT3, NF-κB, and HIF-1α [52], resulting in more cancer-related inflammations leading to the promotion of cell proliferation, cell invasions, angiogenesis, and metastasis. The up-regulation of the inflammatory genes in macrophages and epithelial cells is induced by IL-1β, IFN-γ, TNF-α, and LPS [53]. The previous study showed that treatment of the HT-29 colon cancer cell line with a combination of TNF-α, INF-γ, and LPS has resulted in the up-regulation of pro-inflammatory enzymes (COX-2 and iNOS) and cytokines (IL-1β and TNF-α) expression [54]. On the other hand, the modulation of the inflammatory tumor microenvironment controls the progression of carcinogenesis [41]. In vivo experiments found that most of the inflammation occurred at the systemic level; it might be caused by many immune cells, such as macrophage cells, and then we hypothesized that PCE might inhibit the cancer cell proliferation that is induced by the inflammatory process. Thus, the effect of PCE on colon cell proliferation was studied in inflamed HT-29 and HCT-116, induced by interleukin-6 (IL-6). IL-6 plays a key role in the promotion of cell proliferation and inhibition of apoptosis via its receptor (IL-6Rα) and co-receptor glycoprotein 130 (gp130), resulting in the activation of the JAK/STAT signaling pathway [55]. STAT belongs to a family of transcription factors closely associated with tumorigenic processes, including cancer initiations and progressions [56]. As expected, PCE at the indicated concentrations could inhibit the cell proliferation of IL-6-induced colon cancer cell lines HCT-116 and HT-29 but did not affect stimulated cancer cells. Therefore, the modulation of inflammatory signaling pathways that control their growth needs to be further investigated. Moreover, several reports have shown the application of anti-inflammatory compounds for cancer treatments, including all-trans-retinoic acid (ATRA), vitamin D, non-steroidal anti-inflammatory drugs (NSAIDs), and anti-inflammatory antibodies [42]. For example, ATRA (Vesanoid, Tretinoin) is the primary biologically active metabolite of vitamin A that possesses anti-inflammatory properties. It reduces the expression of immunosuppressive genes, including PD-L1 and IL-10, in advanced-stage melanoma patients [42]. Vitamin D (such as calcitriol) exhibits anti-inflammatory actions that contribute to its beneficial effects on many cancers via inhibition of the synthesis of prostaglandins, suppression of stress-activated kinase signaling, and suppression of NF-κB signaling [42]. Therefore, the mechanisms of inflammatory resolution are of vital importance for cancer prevention. Following in vitro studies, PCE could modulate cytokine secretion in macrophages and could alter the inflammatory microenvironment response of colon cancer cell lines, which was correlated to inflammation in the previous in vivo experiment, which showed a strong correlation between the number of ACF and each cytokine level (R^2^ ˃0.90). These findings could be used to determine the effect of PF seed residue on the progression of rat colonic aberrant crypts induced by DMH and DSS.

Finally, for further application, the human equivalent dose (HED) was converted from the effective dose in rats. In this study, when rats were fed with 1 g/kg body weight of PCE, they received rosmarinic acid (about 21.06 mg/kg body weight), luteolin, and apigenin (about 11.11 and 6.55 mg/kg body weight, respectively). This concentration was equivalent to 9.73 g/day of PCE or 132.5 g of PF seed residue consumption in humans (adults weighing 60 kg), which this ingestion were equal to the receiving of rosmarinic acid (204.9 mg/day), luteolin (108.10 mg/day), and apigenin (63.73 mg/day). Similarly, 9.45 g/day of PF leaves receive rosmarinic acid (about 204.9 mg/day), luteolin, and apigenin (about 0.6 and 1.2 mg/day, respectively). However, the applicable concentration or the formulation of the extract will be further determined.

## 5. Conclusions

The active components in PF seed residue could inhibit the inflammatory microenvironment generated from the activated immune cells and the modulation of gut microbiota, leading to the response suppression of aberrant cells for the tumor-promoting processes. There are limitations to this study. Although we found that PF seed residue suppressed the inflammation-induced ACF progression, the mechanisms of PCE on microbiota related to inflammation and inflammatory-induced colon cancer progression need to be further investigated. However, the consumption of food supplement products made from PF oil, seed residue, or leaves should be a good strategy for colon cancer prevention.

## Figures and Tables

**Figure 1 foods-12-00988-f001:**
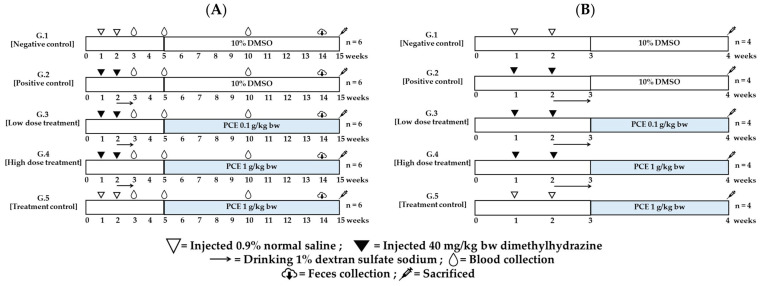
Animal protocol for studying effects of PCE on inflammation induced rat colonic ACF progression (**A**) and short-term colonic inflammation (**B**).

**Figure 2 foods-12-00988-f002:**
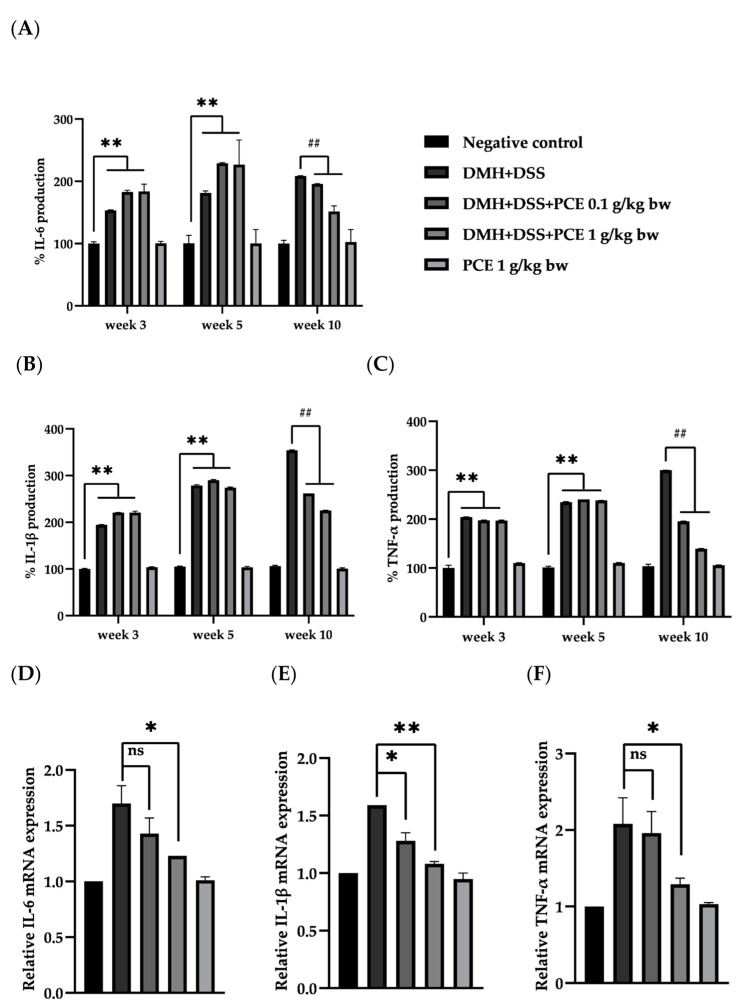
The effect of PCE on cytokines production in rat serum at each time point of blood collection: IL-6 (**A**), IL-1β (**B**), and TNF-α (**C**), with each bar representing the means ± SEM from three rats. ** *p* < 0.01 significant different from negative control, ^##^
*p* < 0.01 significant different from positive control. The fold change of IL-6 (**D**), IL-1β (**E**), and TNF-α (**F**) mRNA in rat colonic epithelial cells, with each bar representing the means ± SD of fold change from four rats. ns: not significant, * *p* < 0.05 and ** *p* < 0.01 significantly different from positive control.

**Figure 3 foods-12-00988-f003:**
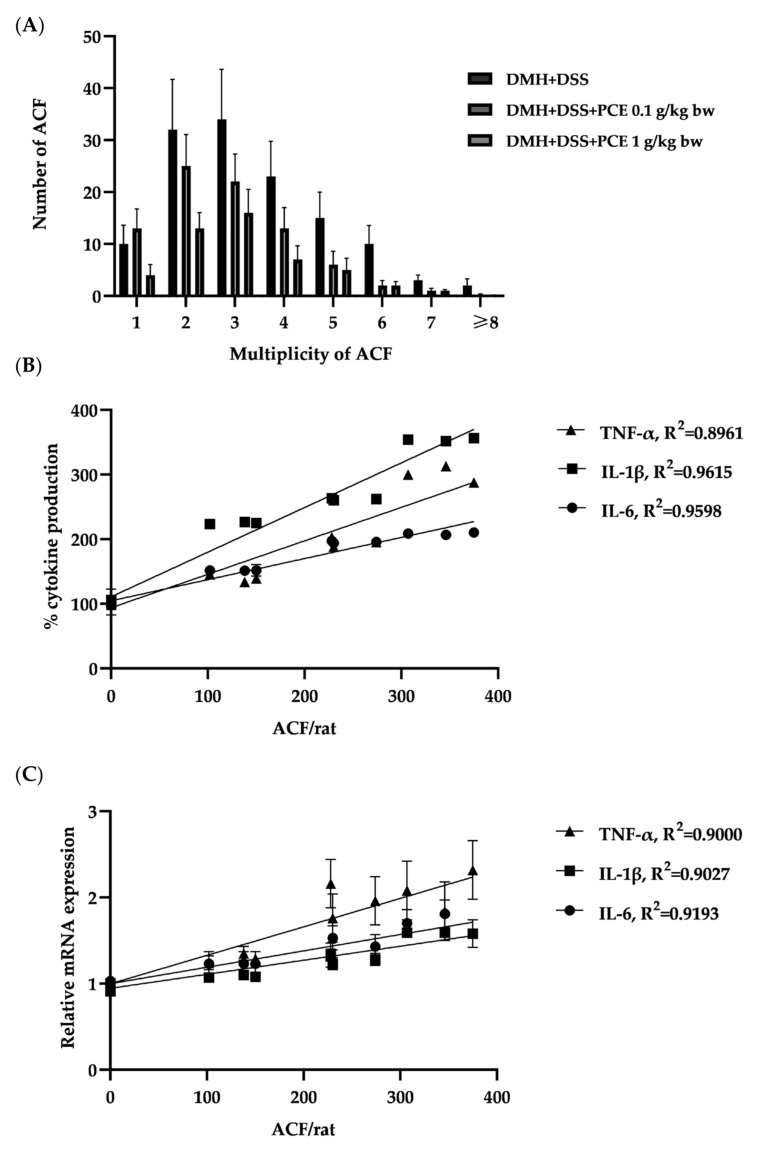
The effect of PCE on number and size distribution of ACF in rat colon (**A**), with each bar representing mean ± SEM of six rats and the correlation between percent cytokine production in serum (**B**) and cytokine expression in colonic epithelial cells (**C**) with mean number of ACF.

**Figure 4 foods-12-00988-f004:**
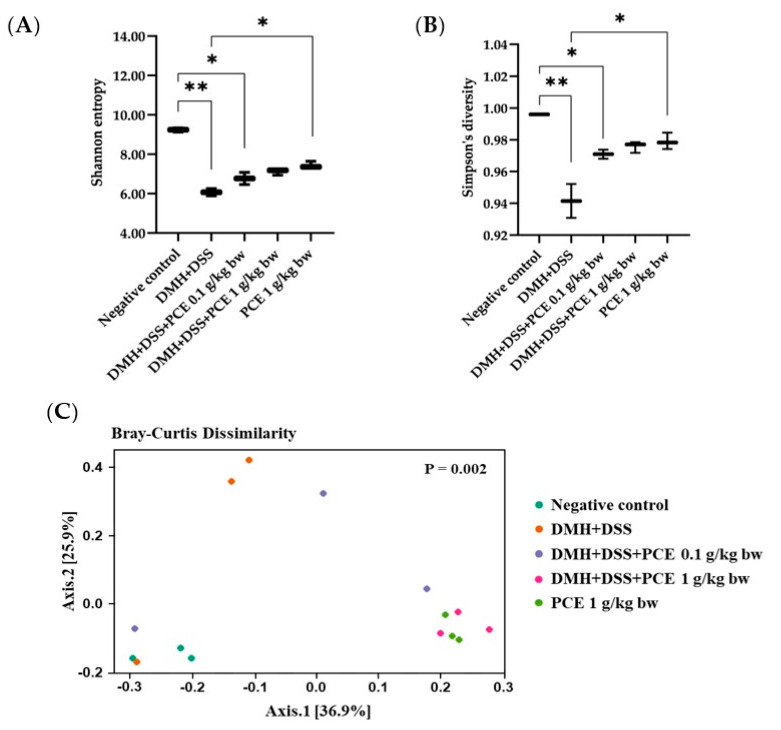
Biodiversity of intestinal microbiota. Alpha diversity indices of richness and evenness using (**A**) Shannon entropy and (**B**) Simpson index. (**C**) Beta diversity index of microbial composition by principal coordinate analysis (PCoA) of Bray-Curtis distance. * *p* < 0.05, ** *p* < 0.01.

**Figure 5 foods-12-00988-f005:**
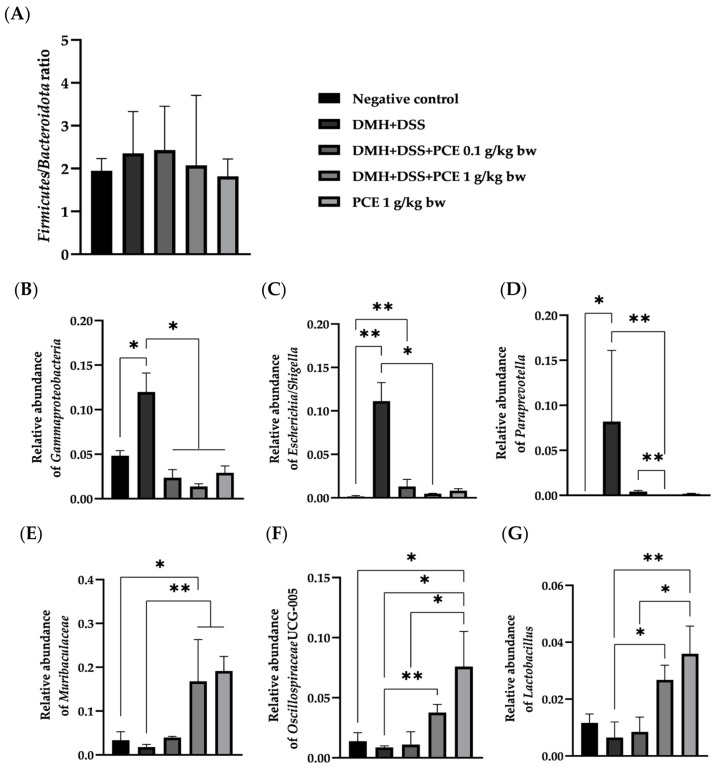
Relative abundance of bacterial biomarkers. (**A**) The ratio of *Firmicutes* to *Bacteroidota* (F/B). Relative abundance of (**B**) *Gammaproteobacteria*, (**C**) *Escherichia-Shigella*, (**D**) *Paraprevotella*, (**E**) *Muribaculaceae*, (**F**) *Oscillospiraceae* UCG-005, and (**G**) *Lactobacillus*. * *p* < 0.05, ** *p* < 0.01.

**Figure 6 foods-12-00988-f006:**
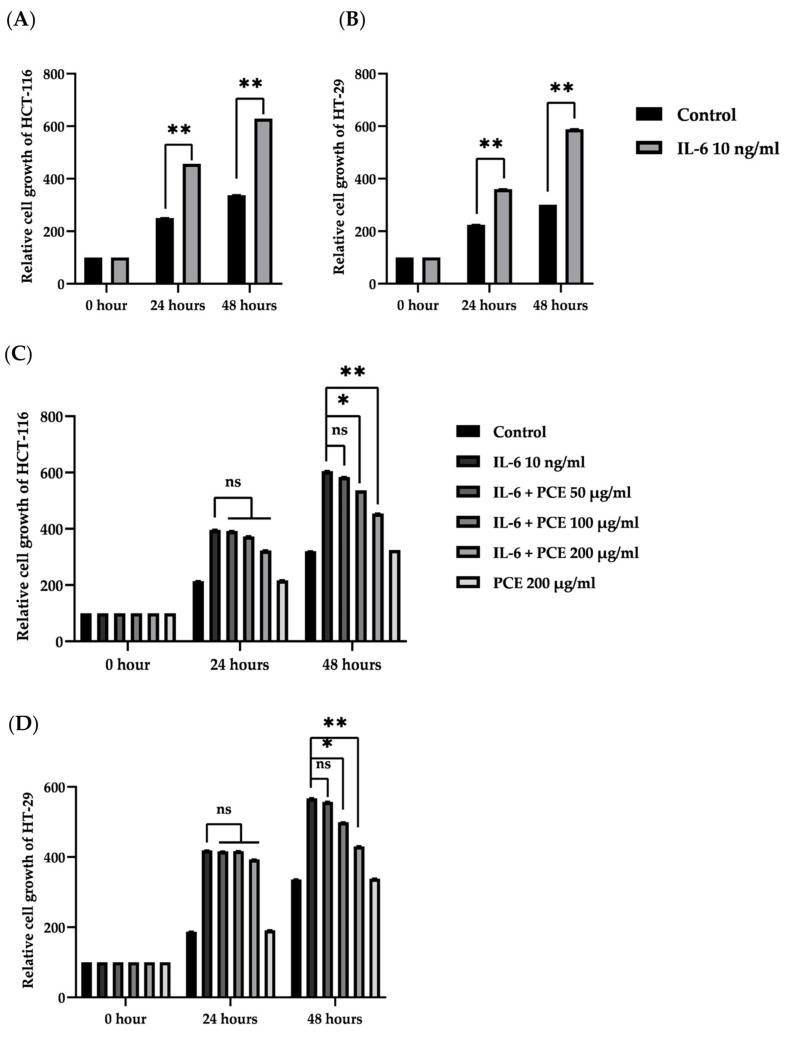
The effect of PCE on 2.5 × 10^3^ cells/well of HCT-116 (**A**) and HT-29 (**B**), which was induced by IL-6 (10 ng/mL) for 24 and 48 h. Each bar represents the Mean ± SD, *n* = 3, ** *p* < 0.01 significant different from control group. The relative cell growth of HCT-116 (**C**) and HT-29 (**D**), treated with IL-6 at 10 ng/mL alone or together with PCE (50, 100, and 200 µg/mL) for 24 and 48 h. Each bar represents the mean ± SD, *n* = 3, ns: not significant, * *p* < 0.05 and ** *p* < 0.01 significantly different from positive control.

**Figure 7 foods-12-00988-f007:**
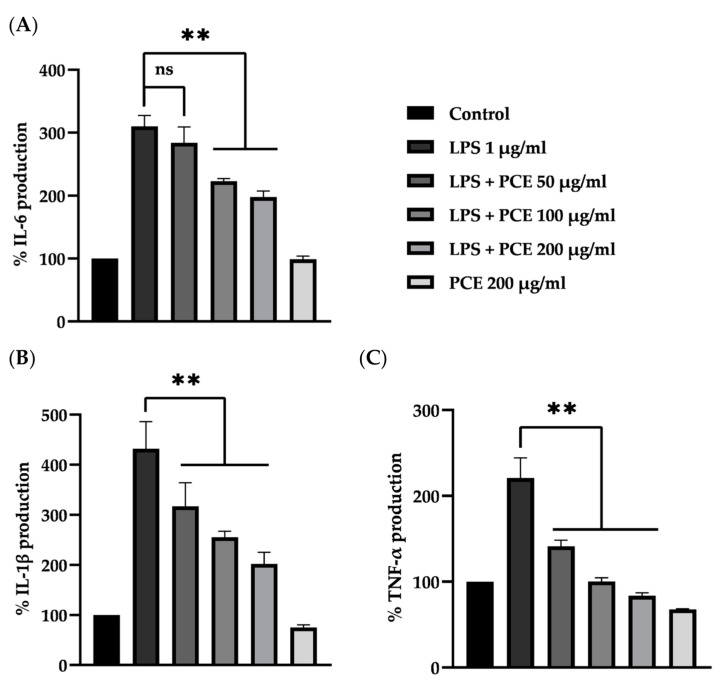
The effect of PCE on the secretion of IL-6 (**A**), IL-1β (**B**), and TNF-α (**C**) in 5 × 10^5^ cells/well of RAW 264.7 cell culture media. Each bar represents the mean ± SD, *n* = 3, ns: not significant, ** *p* < 0.01 significantly different from positive control.

**Table 1 foods-12-00988-t001:** Mean number of ACF and AC/f with the percentage of the inhibition.

Group	*n*	ACF/Rat ^#^			Total Number of ACF ^#^	AC/f ^#^
		Rectum	Proximal	Distal		
**Negative control**	6	0 ± 0.00	0 ± 0.00	0 ± 0.00	0 ± 0.00	0 ± 0.00
**DMH + DSS**	6	28 ± 10.23	201 ± 23.07	159 ± 14.63	388 ± 90.11	3.31 ± 0.41
**DMH + DSS + PCE** **0.1 g/kg bw**	6	19 ± 11.96 * (33.93%)	115 ± 29.91 ** (42.47%)	117 ± 7.63 * (26.26%)	251 ± 56.53 ** (35.20%)	2.81 ± 0.40 ^NS^ (15.12%)
**DMH + DSS + PCE** **1 g/kg bw**	6	11 ± 9.29 * (59.82%)	44 ± 29.45 ** (77.07%)	75 ± 11.28 ** (53.02%)	130 ± 31.78 ** (66.46%)	3.09 ± 0.27 ^NS^ (6.69%)
**PCE 1 g/kg bw**	6	0 ± 0.00	0 ± 0.00	0 ± 0.00	0 ± 0.00	0 ± 0.00

^#^ Mean ± SD of 6 rats in each group. ^NS^ not significant, * *p* < 0.05 and ** *p* < 0.01 compared to positive control.

## Data Availability

The datasets generated for this study are available on request to the corresponding author.

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
