# Peer review of "The Extract of Perilla frutescens Seed Residue Attenuated the Progression of Aberrant Crypt Foci in Rat Colon by Reducing Inflammatory Processes and Altered Gut Microbiota"

_foods, 2023, doi:10.3390/foods12050988_

Round 1
Reviewer 1 Report
Please, see all comments in the file attached.

Author Response
We are very grateful to the editors and reviewers for your serious and responsible review to our manuscript and thank you very much for giving us valuable comments and suggestions, which have important guiding significance for improving our writing level and future scientific research. We tried to response point by point according to the editor's and reviewers' comments and highlighted with yellow to indicate the editing points in the revised manuscript.
Comments from Reviewer 1
General concept comments
1) Particularly, in the title the authors claim that “Perilla frutescens seed residue attenuated the progression of aberrant crypt foci in rat colon by reducing inflammatory processes related to gut microbiota”. But the authors only have found changes in taxonomic composition of colon microbiota in few experimental groups. They did not study relationship between gut microbiota and colon inflammation. Moreover, they state (lines 406-407) that “…the cause-effect relationships of these bacteria in CRC pathogenesis are still poorly understood [41].”. If it is so, the title should be rephrased.
Response
Thank you for your informative comment. Firstly, we agree with the reviewer that we have not directly determined the relation of the changes of gut microbiota and colonic inflammation in this study. Therefore, the title was changed to “The extract of Perilla frutescens seed residue attenuated the progression of aberrant crypt foci in rat colon by reducing inflammatory processes and altered gut microbiota”.
However, we tried to link between the increasing of Gammaproteobacteria and Escherichia/Shigella which are related to colonic inflammation and colorectal cancer (line 393-401). On the other hand, PCE consumption reduced Gammaproteobacteria levels related to attenuation of the chronic inflammatory condition in the gut of DMH + DSS rats. Moreover, the beneficial bacteria are strongly showed the anti-cancer activity such as Lactobacillus and Oscillospiraceae (line 404-408) with new references. Therefore, the discussion in this part was modified as following (line 395 – 413).
The discussion was edited as following;
Our finding showed that PCE increased the gut microbiota diversity in the CRC rat model. Moreover, there was an increase in the relative abundance of Gammaproteobacteria, an important class of bacteria in the phylum Pseudomonadota (formerly named Proteobacteria) in DMH + DSS treated rats. High numbers of Proteobacteria have been considered an important marker for gut dysbiosis in several medical conditions, including colorectal cancer [44]. A lipopolysaccharide outer membrane of Gammaproteobacteria causes chronic inflammation through Toll-like receptor (TLR)-4 activation in the gut, which can be modulated by some bioactive compounds [45]. Interestingly, PCE reduced the increase of Escherichia/Shigella, the representative bacteria belonging to a class of Gammaproteobacteria. These data suggested that PCE consumption might have the ability to attenuate the chronic inflammatory condition in the gut of DMH + DSS rats and reduce Gammaproteobacteria levels. Treatment of PCE also increased the relative abundance of beneficial bacteria such as Muribaculaceae, Lactobacillus and Oscillospiraceae. The protective effects of Lactobacillus on the development of precancerous growths and colorectal carcinogenesis in the rat model have been revealed [46]. Moreover Ruminococcaceae (Oscillospiraceae) is a family of strictly anaerobic bacteria that are normally present in the colonic mucosal biofilm of healthy individuals [47] However, the interaction of gut microbiota, immune cells, inflammatory cytokines, and the response of aberrant colonic epithelial cells are the focus point for prevention of inflammation-related colon carcinogenesis
[46] Sivamaruthi, B.S., P. Kesika, and C. Chaiyasut, The Role of Probiotics in Colorectal Cancer Management. Evid Based Com-plement Alternat Med, 2020. 2020: p. 3535982.
[47] DeWeirdt, R.; Van DeWiele, T. Micromanagement in the gut: Microenvironmental factors govern colon mucosal biofilm structure and functionality. npj Biofilms Microbiomes 2015, 1, 15026
2) In the study aim (lines 56-60) the authors indicated that they studied effects of PCE to “…the promotion stage of rat colon carcinogenesis, which was induced by chronic inflammation and alteration of gut microbiome.” But, frankly speaking, it is impossible to state that aberrant crypt foci (ACF) in the studied rat guts were caused by chronic inflammation and alteration of gut microbiome only. First, we cannot exclude other numerous possible reasons of ACF, such as mutation rate, colonocytes dystrophy produced by dextran and so on. Second, a role of gut microbiome as a cause of colon carcinogenesis is discussed now, but has not been proved reliably yet. Moreover, in the study the authors have observed the changes in gut microbiome composition, but their relationship with carcinogenesis has not been proved also. Thus, I recommend to modify the study aim as follow “…the promotion stage of rat colon carcinogenesis, general and local gut inflammation, and alteration of gut microbiome.”
Response:
Thank you for your kind suggestion, the aims of this study were edited according to your suggestion in Line 59-63.
“This study aimed to investigate the inhibitory activities of ethanolic extract of Perilla frutescens seed residue on the promotion stage of rat colon carcinogenesis, general and local gut inflammation, and alteration of gut microbiome. Moreover, the molecular mechanisms of PCE on inflammatory responses in both macrophage and human colon cancer cell lines were also determined.”
Specific comments
1) Line 101. Misprint, miss PCE? “…PCE 0.1 or 1 g/kg body weight/day…”
Response: PCE was added. (Line 104)
2) Line 137. Here and below incorrect references are used (PMID: 21702898).
Response: The references were already checked and sequenced. (Line 137 and below)
3) Line 216. “The levels…” would be better than “The results…”.
Response: The sentence was edited. (Line 216)
4) Line 241. “…in colonic epithelial cells of untreated animals.” would be better than “…in colonic epithelial cells.”.
Response The sentence was edited. (Line 241)
5) Line 270. Here and below all Latin names of bacteria should be written in Italian, e.g. “Escherichia/Shigella”.
Response: These Latin names were italic formatted.
6) Lines 259-262. Based on the Figure 4C the statement that “Interestingly, rats fed with PCE had changed their gut microbiota compared to the untreated group (beta diversity with Bray-Curtis dissimilarity, Figure 4C).” is not correct. In the picture dots designating rats fed with PCE 0.1 g/kg body weight are clustered with both negative control and DMH+DSS animals. Only dots of rats fed with PCE 1 g/kg including treatment control and DMH+DSS+PCE 1 g/kg form a cluster exactly remote from both negative control and DMH+DSS animals. And of course, rats could not change their gut microbiota themselves. Please, rephrase the statement.
Response: We agreed with the reviewer’s comment to point out the dose effect of PCE on rat’s gut microbiota beta-diversity in Figure 4C. We have already rephrased these sentences in the revised manuscript. Now, it reads “Interestingly, the rat’s gut microbiota beta diversity with Bray-Curtis dissimilarity could be observed only in rats fed with the high dose of PCE (1 g/kg rat body weight), not a low dose (0.1 g/kg rat body weight).” (Line 250-263)
7) Section 3.6. The MTT assay which is described in proper section 2.11 has been used by authors to estimate a value of relative cell proliferation, but not “relative cell viability” or “relative cell numbers”. These terms have absolutely different meanings. Please use in all parts of the manuscript only one term that was defined in section 2.11 .
Response: The words were changed to relative cell growth in both section 2.11, and 3.6 which also shown in figure 6.
In section 2.11, the sentences was changed to …. Cell viability was assessed by the MTT test at the set period compared to 100% cell viability at 0 hour and calculated as relative cell growth. (Line 175-176)
8) Line 314. Misprint? Did the authors mean “Inflammatory process”?
Response: This point is already corrected. (Line 316)
9) Line 315-316. Double phrase. “by reducing the cytokines production in LPS-induced RAW 264.7 cell. by reducing the cytokines production in LPS-induced RAW 264.7 cell.”
Response: The repeated phrase was removed.
10) Lines 358-361. The phrase is too long, contains three subjects and predicates, and is not clear. It would be better to rephrase it and divide into two sentences.
Response: The sentence was rephrased and separated to 2 sentences.
It has been reported that DMH-induces genetic alterations of colonic epithelial cells and ACF is formed. Then DSS treatment alters the colonic mucosal inflammation from various cytokines leading to ACF progression. (Line 360-363)

Reviewer 2 Report
This study investigated the inhibitory effect of PCE, an ethanolic extract of Perilla seed residue, on the promotion phase of enteritis-associated colon cancer in rats induced by chronic inflammation and intestinal flora alteration, as well as the molecular mechanism of PCE in activating macrophage and inflammatory responses in colon cancer cell lines, which provides a new direction for the study of colon cancer prevention. Nevertheless, there are some problems with the article.
1. This study used DMH and DSS to prepare an animal model of enteritis-associated colon cancer in rats, but the description of the evaluation criteria of this model is missing, please add.
2. In the investigation of the inhibitory effect of PCE on DMH and DSS-induced ACF in rats, the concentrations of PCE chosen were 0.1 and 1 g/kg, please explain the reasons for choosing these two concentrations.
3. In the animal protocol to investigate the effect of PCE on ACF induced by DMH and DSS in rats, the intervention for group 5 rats does not match the description in Figure 1, please correct it. In addition, please ask the authors to explain the role of setting up group 5 rats in the article so that readers can better understand the article.
Author Response
We are very grateful to the editors and reviewers for your serious and responsible review to our manuscript and thank you very much for giving us valuable comments and suggestions, which have important guiding significance for improving our writing level and future scientific research. We tried to response point by point according to the editor's and reviewers' comments and highlighted with yellow to indicate the editing points in the revised manuscript.
Comments from Reviewer 2
This study investigated the inhibitory effect of PCE, an ethanolic extract of Perilla seed residue, on the promotion phase of enteritis-associated colon cancer in rats induced by chronic inflammation and intestinal flora alteration, as well as the molecular mechanism of PCE in activating macrophage and inflammatory responses in colon cancer cell lines, which provides a new direction for the study of colon cancer prevention. Nevertheless, there are some problems with the article.
Responses to reviewer 2:
- This study used DMH and DSS to prepare an animal model of enteritis-associated colon cancer in rats, but the description of the evaluation criteria of this model is missing, please add.
Response: The evaluation criteria of this model is ACF formation induced by DMH (number of ACF), according to the criteria of Bird RP, whereas the large AC/f is used to evaluate the progression of ACF related to inflammation induced by DSS.
Therefore, the sentences of ACF criteria are added.
“Under a light microscope, the quantity and size of ACF were graded in accordance with Bird RP criteria [23,24]. Compared to normal crypts, aberrant crypts were bigger and had a thicker epithelial lining, and usually gathered into a focus of small (1-3 AC/f) or clusters of abnormally large crypts (>4 AC/f).” (Line 122-126)
- In the investigation of the inhibitory effect of PCE on DMH and DSS-induced ACF in rats, the concentrations of PCE chosen were 0.1 and 1 g/kg, please explain the reasons for choosing these two concentrations.
Response: Our previous studies showed that the crude ethanolic extract of PF leaves exhibited anti-inflammation in vitro and inhibitory effects on DMH-induced ACF formation in rats. This extract contained 7 times higher of rosmarinic acid but 12 and 16 times lower of luteolin and apigenin, respectively when compared to PCE. Therefore, we designed the high dose (1 mg/Kg bodyweight) which represented rosmarinic acid and leuteolin about 21.06 mg 11.11 mg per kg bodyweight. These doses are higher than the doses in the previous reports which showed the preventive effect against carcinogen induced ACF formation in rodent [1, 2]
- Furtado RA, Oliveira BR, Silva LR, Cleto SS, Munari CC, Cunha WR, Tavares DC. Chemopreventive effects of rosmarinic acid on rat colon carcinogenesis. Eur J Cancer Prev. 2015 Mar;24(2):106-12. doi: 10.1097/CEJ.0000000000000055. PMID: 24977626.
- Pandurangan AK, Ananda Sadagopan SK, Dharmalingam P, Ganapasam S. Luteolin, a bioflavonoid, attenuates azoxymethane-induced effects on mitochondrial enzymes in BALB/c mice. Asian Pac J Cancer Prev. 2014 Jan;14(11):6669-72. doi: 10.7314/apjcp.2013.14.11.6669. PMID: 24377586.
This point was discussed in line 337-340;
The effective dose of PCE was referred to our previous research which suggests that crude ethanolic extract of PF leaves containing high rosmarinic acid exhibited anti-inflammation in vitro and inhibitory effects on DMH-induced ACF formation in rats in the promotion stage [16].
- In the animal protocol to investigate the effect of PCE on ACF induced by DMH and DSS in rats, the intervention for group 5 rats does not match the description in Figure 1, please correct it. In addition, please ask the authors to explain the role of setting up group 5 rats in the article so that readers can better understand the article.
Response: the description is already corrected.
“The setting up of group 5 rats for PCE control at truly high dose (1 g/ kg body weight) to make sure that PCE is not alter the inflammation of colonic mucosal cells and not induce the ACF formation. Moreover, we would like to compare the gut microbiota in PCE treated rats (without the aberration of colonic mucosal cells) to normal rats.”

Reviewer 3 Report
I have a major concern on the age of the animals and the limitations of the study were not stated in the manuscript. The young animals might produce the results where wouldn't ( no evidence in the manuscript).
The conclusion is overestimated, and the limitations are missing in this section.
Four-week-old male Wistar rats (60-90 g) were used, which needs to be justified. These are still babies and should be young adults at around 8 weeks old. The low weight might overestimate the results. Please justify why you have used such young animals.
The abstract is incomplete and it must me more organised. It should include all the items Introduction, results, discussion, conclusions and limitations.
Author Response
We are very grateful to the editors and reviewers for your serious and responsible review to our manuscript and thank you very much for giving us valuable comments and suggestions, which have important guiding significance for improving our writing level and future scientific research. We tried to response point by point according to the editor's and reviewers' comments and highlighted with yellow to indicate the editing points in the revised manuscript.
Responses to Reviewer 3:
- I have a major concern on the age of the animals and the limitations of the study were not stated in the manuscript. The young animals might produce the results where wouldn't (no evidence in the manuscript).
Response:
- Although 4-week-old rats were order from animal provider, after transportation, they were acclimatized for at least 1 week before the procedure began. Therefore, the beginning of induction rats were more than 5 weeks old with average weight about 111.87±2.09 grams. In our experience and several reports, rats at 5-week old response to 2 doses of DMH or AOM within 2 weeks to initiate the ACF formation. Moreover, during DSS treatment (at week 3 in protocol) to generate the inflammation, rats were 7-week-old which represented exactly at the adult stage. In our results can confirm that rats received DMH+DSS (positive control) showed comparable ACF formation and inflammation.
However, we added the average weight of rats used to initiate this DMH+DSS treatment model on the first sentence of treatment protocol (line 96-97).
“After 1 week acclimatization, rats were randomly separated into five groups of six rats with comparable average weights (111.87±2.09 g)”
- The conclusion is overestimated, and the limitations are missing in this section.
Four-week-old male Wistar rats (60-90 g) were used, which needs to be justified. These are still babies and should be young adults at around 8 weeks old. The low weight might overestimate the results. Please justify why you have used such young animals.
Response:
Although we have tried to claim that PCE administration can reduced the inflammation process leading to suppress the progression of ACF and also alter gut microbiota. The relation of gut microbiota and inflammation process was not clear. Moreover, the molecular mechanisms of anti-inflammation and cytokine responses were studies only in cell culture model. The mechanisms underlining the effect of PCE on inflammation related to gut microbiota alteration and regulation of aberrant cells progression need to be further investigated. Therefore, the conclusion was edited.
In conclusion, the limitation was added … “There are limitation in this study. Although we found that PF seed residue suppressed the inflammation-induced ACF progression, the mechanisms of PCE on microbiota which related to inflammation and inflammatory-induced colon cancer progression needs to be further investigate.”
- The abstract is incomplete, and it must me more organized. It should include all the items Introduction, results, discussion, conclusions, and limitations.
Response: The abstract was reorganized and added the limitation of this study, as following;
Abstract: Perilla frutescens (PF) seed residue is a waste from perilla oil production which still contains nutrients and phytochemicals. This study aimed to investigate the chemoprotective action of PF seed residue crude ethanolic extract (PCE) on the inflammatory induced promotion stage of rat colon carcinogenesis and cell culture models. PCE 0.1 and 1 g/kg body weight were oral gavage to rats after receiving dimethylhydrazine (DMH) with one week of dextran sulfate sodium (DSS) supplementation. PCE at high dose exhibited a reduction of aberrant crypt foci (ACF) number (66.46%) and decreased pro-inflammatory cytokines compared to DMH+DSS group (P<0.01). Additionally, PCE could either modulate the inflammation induced murine macrophage cell by bacterial toxin or suppress the proliferation of cancer cell lines which was induced by the inflammatory process. These results demonstrate that the active components in PF seed residue showed a preventive effect on the aberrant colonic epithelial cell progression by modulating inflammatory microenvironments from the infiltrated macrophage or inflammatory response of aberrant cells. Moreover, consumption of PCE could alter rat microbiota which might be related to health benefit. However, the mechanisms of PCE on microbiota which related to inflammation and inflammatory-induced colon cancer progression needs to be further investigate.
